# A code for clinical trials centralized monitoring, *sharing open-science solutions to high-quality data*

**André Daher**[1], **Júlio Castro-Alves**[2], **Leandro Amparo**[1], **Natalia Pacheco de Moraes**[1], **Thaís Regina Araújo dos Santos**[1], **Karla Regina Gram dos Santos**[1], **Cristiane Siqueira do Valle**[1], **Maria Hermoso**[1]*, **Margareth Catoia Varela**[2], **Rodrigo Correa Oliveira**[1]

**1** Vice-presidency of Research and Biological Collections, Oswaldo Cruz Foundation (Fiocruz), Rio de Janeiro, Brazil, **2** National Institute of Infectious Disease, Oswaldo Cruz Foundation (Fiocruz), Rio de Janeiro, Brazil

* maria.hermoso@fiocruz.br

**Data Availability Statement:** The tool R code and the REDCap project (PHP script, CRFs and aditional

## Abstract

Monitoring of clinical trials is critical to the protection of human subjects and the conduct of high-quality research. Even though the adoption of risk-based monitoring (RBM) has been suggested for many years, the RBM approach has been less widespread than expected. Centralized monitoring is one of the RMB pillars, together with remote-site monitoring visits, reduced Source Data Verification (SDV) and Source Document Reviews (SDR). The COVID-19 pandemic promoted disruptions in the conduct of clinical trials, as on-site monitoring visits were adjourned. In this context, the transition to RBM by all actors involved in clinical trials has been encouraged. In order to ensure the highest quality of data within a COVID-19 clinical trial, a centralized monitoring tool alongside Case Report Forms (CRFs) and synchronous automated routines were developed at the clinical research platform, Fiocruz, Brazilian Ministry of Health. This paper describes how these tools were developed, their features, advantages, and limitations. The software codes, and the CRFs are available at the Fiocruz Data Repository for Research—Arca Dados, reaffirming Fiocruz's commitment to Open Science practices.

## Introduction

| Statement of significance | |
|---|---|
| Problem | Risk-based monitoring (RBM) approaches for clinical trials are less widespread than expected. |
| What is Already Known | Although recommended by regulatory agencies and ICH guidelines, the lack of technologies for RBM is one of the reasons for the slow adoption of RBM practices |
| What This Paper Adds | This study aims to describe a tool developed for central monitoring of a COVID-19 clinical trial alongside Case Report Forms and synchronous automated routines. Additionally, the code for the tool is shared, which is worthwhile to contribute to open science and boost access to technologies for easier implementation of RBM practices in clinical trials. |

files) are available at the data sharing platform ARCA Dados/Fiocruz: https://arcadados.fiocruz.br/dataset.xhtml?persistentId=doi:10.35078/QCXI6N.

**Funding:** The Vice-presidency of Research and Biological Collections, Oswaldo Cruz Foundation (Fiocruz), Ministry of Health of Brazil provided the funding. The funder had no role in study design, data collection and analysis, decision to publish, or preparation of the manuscript.

**Competing interests:** The authors declare no competing interests.

The COVID-19 pandemic has highlighted that clinical trials conducted in accordance with good clinical practices are needed to support evidence-based policies with high-quality data while protecting the safety, well-being, and rights of the participants. Monitoring of clinical trials is critical to the protection of human subjects and the conduct of high-quality research. Thus, the COVID-19 pandemic produced a paradox: whilst the high-quality data was most needed, the conduction of clinical trials on-site monitoring visits were adjourned [1] as self-isolation, travel bans, social distance and lockdowns were mandatory to prevent the spread of the virus. This was the essential drive to implement risk-based monitoring (RBM) in many clinical trials.

Even though the adoption of risk-based monitoring has been recommended by regulatory agencies since 2013 [2,3] and it is also suggested by the ICH guidelines in their last 2016 revision [4] the RBM approach has been less widespread than expected. The slow RBM adoption was due to the former complexity of implementing these practices, the need for new technologies, an incorrect assumption that RBM methodology data are less likely to satisfy regulators [5] and the lack of familiarity with RBM.

In short, risk-based monitoring comprises of remote-site monitoring visits and centralized monitoring; besides reduced Source Data Verification and Source Document Reviews.

Source Data Verification (SDV) of all case report forms (CRF) (100% SDV) may avert systematic errors at the initial monitoring visits to clinical sites. All along the study, a complete verification of a fixed set of essential documents and data, such as informed consent, serious adverse events (SAE), drug accountability and primary outcomes measurements, may be needed to avert fraud, and ensure data integrity. Though, reducing the amount of SDV in favour of a more targeted approach, i.e., reduced SDV and SDR alongside the modern data-cleaning statistical approaches might be adequate to ensure the overall data quality [6], improving the efficiency, timelines, and costs of the clinical trials oversight process [7], including COVID trials [8]. Remote-site monitoring with reduced SDV was a *sine qua non* condition during the pre-vaccine period of the pandemic [9].

Centralized monitoring consists of remotely assessing the quality of electronic data to identify data discrepancies through automated assessments. It encompasses data review and analysis performed remotely from the trial site to scrutinize the collected subjects' data to identify unusual patterns, protocol deviations or missing or invalid data [10]. Electronic data capture systems that are secure and provide audit trails for tracking data manipulation, such as the electronic case report form (eCRF) REDCap [11], have made one of the pillars of the RBM, centralized monitoring, a feasible task. Still, the lack of availability of open-source software, tools, detailed guidance, and challenges in operationalization [12] have hindered centralized monitoring adoption.

Herein we share the code of a centralized monitoring tool alongside CRFs and synchronous automated routines that were developed at the clinical research platform, Fiocruz, Brazilian Ministry of Health for a COVID-19 clinical trial.

This centralized monitoring tool remotely reviews the quality of electronic clinical data to identify protocol deviations, pharmacovigilance alert signs, study performance metrics and data completeness through automated assessments. Using an R script to assure the reproducibility of this data quality assessments, the tool might enhance subjects' safety, as the control of AE is improved; accelerate data cleaning before statistical analysis; evaluate performance metrics of trial sites; and decrease the number of on-site monitoring visits, which has an impact on trial costs, timelines, and subjects' safety. Additionally, to contribute to data quality assurance, synchronous automated routines were also built in PHP programming language to automatically transfer data between the different databases. Thereby, core data entry is improved, ensuring 100% of precision and consistency in critical data, such as randomization.

These open-science solutions are powerful tools for data management, pharmacovigilance, and clinical teams involved in clinical trials. Together with other open science initiatives, these tools may help push toward the adoption of centralized monitoring and RBM. Centralized monitoring is a layer within data quality assurance that might be regarded as a mandatory quality standard in clinical trials in a near future.

The objective of this paper is to describe and share these open science solutions. Fiocruz is reaffirming its commitment to Open-Science practices by encouraging the availability of data and information at each stage of the research process. The codes for these tools are available at ARCA Dados (doi:10.35078/QCXI6N), an important element of the Fiocruz-Brazilian Ministry of Health policy for the management and sharing of research data.

## Methods

### Study and study site

These open-science solutions for a COVID-19 clinical trial were developed at the clinical research platform, Fiocruz, Brazilian Ministry of Health. Fiocruz clinical research platform funds and provides support to clinical research that evaluates new technologies to the Brazilian National Health System (SUS). Support includes clinical research crosscutting activities, such as monitoring, data management, ethical and regulatory affairs, and pharmacovigilance. The platform was the coordinator of the trial: *Efficacy and safety evaluation of Favipiravir for treatment of COVID-19*: *an adaptive, multicentre, double-blind, randomised, placebo-controlled clinical trial.* In brief, it was a clinical trial to repurpose favipiravir, an antiviral drug registered in Japan to treat influenza [13], as a treatment to reduce the rate of SARS-CoV-2 infection progression to severe disease. High-risk patients [14] presenting mild to moderate clinical profile of COVID-19 [15] meeting all the inclusion criteria were enrolled and treated for 10 days in this trial. The trial was performed in six clinical trial sites in Brazil, located in 5 different states (Rio de Janeiro, São Paulo, Minas Gerais, Rondônia, and Mato Grosso do Sul). Further study details are available at the Brazilian Clinical Trials Registry (RBR-85vh8fx), a primary WHO repository.

### The databases

The REDCap [16] (version 12.0.28), an electronic data capture system available free of charge for non-profit research organizations, was used to this trial data management.

The CRFs were designed to allow automatic statistical analysis. Whenever possible, open text fields were replaced by closed questions reducing misunderstandings and time to answer. These closed questions were presented in three different multiple-choice field types: checkboxes, to select multiple answers at the same time; dropdown lists; and radio buttons, to select a single answer within mutually exclusive options. Further on, these questions were analysed as categorical variables. Moreover, calculated fields were used to perform the automatic computation of two or more fields, such as age calculation or Body Mass Index. Text Boxes were designed with field validation to ensure quality. Branching logic guaranteed that conditional fields just pop up depending on the previous answer, for instance only females could have a positive pregnancy response. When relevant, some of these CRFs were developed according to CDISC (Clinical Data Interchange Standards Consortium) standards, including Concomitant Medications and Adverse Events forms.

Within the REDCap system, the study was divided into four projects, each one with its respective databases. Initial data were entered in a Recruitment/Inclusion project. It included the data from all recruited patients, their eligibility or recruitment failure and when applicable the randomization and the numbers of labels of the selected bottles (either placebo or test

drug). The main project recorded the follow-up data of participants enrolled in the trial. These were the two main project databases. The third project managed the drug bottles' accountability. The last project was the participant´s diary with self-answered questionnaires linked to a reminder message (SMS) operator.

## The tools

To implement a centralized monitoring routine, an application was developed using the R (version 4.0.2) statistical software to write the code and the Shiny R package, used to build webpages based on R codes. This provided web-based data interactive monitoring dashboards through a web server with the Shiny server (version 1.5.13.995). Both the Shiny package for R and the Shiny server are open-source software and are developed and supported by RStudio. (Rstudio Team (2020). Rstudio: Integrated Development for R. Rstudio, PBC, Boston, MA URL http://www.rstudio.com/).

This application accessed and analysed the project databases designed in REDCap [16]. The application is connected to REDCap using an application programming interface (API) token, which is only known to the REDCap administrator and the application developer.

The analysis by R/Shiny was done using approximately 195 fields from the CRFs of the two main databases, which corresponds to almost 38.46% of the total fields designed to capture data for this project. The fields accessed by the Central Monitoring were related to Participant Inclusion (14), Recruitment Failure (1), Inclusion/Exclusion Criteria (14), Adverse Events (23), Laboratorial alterations (48), Schedule adherence (2), Queries (2), Inconsistencies in dates and numerical data (57), Completeness of the CRF fields (32), and Recruitment and Inclusion Rate (2).

In addition to the R script, two scripts in PHP programming language were developed to carry out the automatic export and import of data between the specified projects. The aim of these PHP scripts was to avoid errors in the distribution of medication/placebo bottles and mitigate the risk of including patients in the non-assigned arm causing unbalanced randomization and biased results.

These two scripts in PHP used a combination of two REDCap features to perform a synchronous automated replication; an API that can be used to programmatically retrieve or modify data within REDCap and Data Entry Trigger. This is an advanced feature, its purpose is to execute actions by a remote website, such as making a call to the REDCap API, via HTTP Post request.

These scripts linked the Recruitment/Inclusion and the bottles accountability project selecting the numbers of the labels to be used according to the patient ID at the randomization list and blocking the future use at the drug accountability project. This project had a pre-filled database with all numbers assigned to each bottle available. These values were used to feed the Structured Query Language (SQL) fields on drugs bottle selections at Recruitment/Inclusion project. Additionally, these scripts automatically save the patient ID across the two other projects ensuring a perfect linkage. These two PHP scripts that perform synchronous automated replication had been used for other clinical trials management purposes such as scheduling patients visits.

These codes and the trial CRFs are available at the data-sharing platform ARCA Dados/Fiocruz (doi:10.35078/QCXI6N) upon request. These files are licensed under a Creative commons Attribution (CC BY).

## Ethics statement

This study did not involve actual human participants' data. The clinical trial "*Efficacy and safety evaluation of Favipiravir for treatment of COVID-19*: *an adaptive*, *multicentre*, *double-*

*blind, randomised, placebo-controlled clinical trial"* was approved by the Instituto Oswaldo Cruz Institutional Review Board, CAAE number 46417321.6.1001.5248. Written consent was obtained from all study participants.

## Results

### Results display

An actual dashboard linked to test databases can be accessed at https://shiny.fiocruz.br/teste/pce0121/. Most of the critical central monitoring aspects are evaluated. The rational of parameters election were based on the most frequent monitoring findings, i.e. protocol deviations; data robustness and integrity (completeness of outcomes); and high risk aspects, as pharmacovigilance. Additionally, study sites performance metrics were included as alert signs. Herein a brief description of the parameters measured, their features and their applicability in clinical trial assessments are presented, alongside illustrative print screens.

The initial page of the application presents a brief tutorial on how to use the centralized monitoring and a description of each of the ten tabs: patient inclusion rates, recruitment failure, missing data, inclusion and exclusion criteria, adverse events, laboratorial abnormalities, data validation, follow-up visits according to the study schedule and queries.

## Recruitment and Inclusion Rate

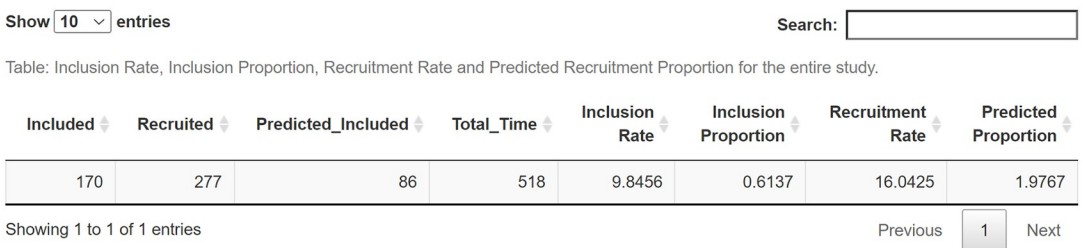

Show 10 ∨ entries                                              Search: [          ]

Table: Inclusion Rate, Inclusion Proportion, Recruitment Rate and Predicted Recruitment Proportion for the entire study.

| Included | Recruited | Predicted_Included | Total_Time | Inclusion Rate | Inclusion Proportion | Recruitment Rate | Predicted Proportion |
|---|---|---|---|---|---|---|---|
| 170 | 277 | 86 | 518 | 9.8456 | 0.6137 | 16.0425 | 1.9767 |

Showing 1 to 1 of 1 entries                              Previous  [1]  Next

Figure: Expected (black line) and actual (gray line) total number of participants throughout the study until now.

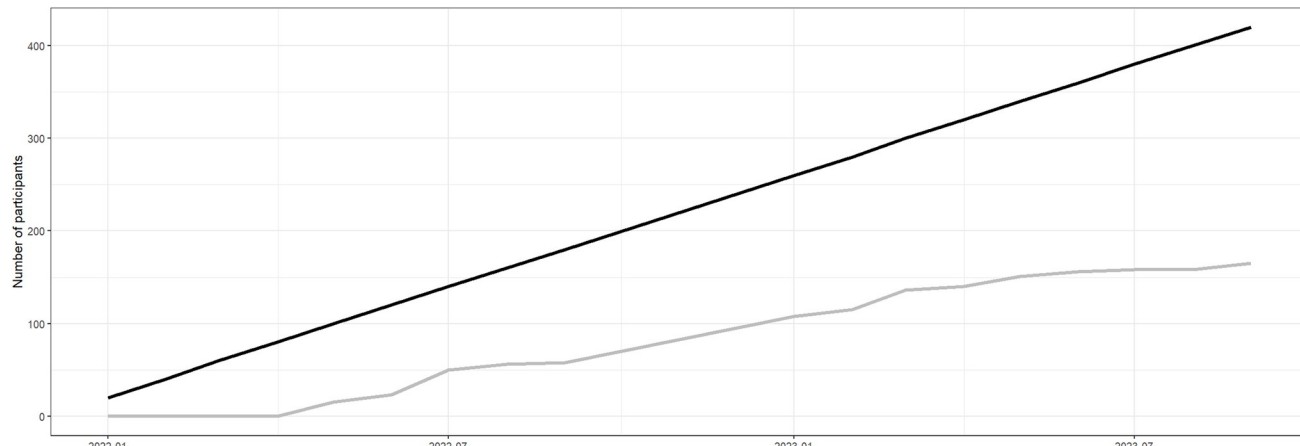

**Fig 1. Subjects' recruitment vs. inclusion enrolment rates overall.**

## Recruitment and Inclusion Rate

Show 10 ∨ entries                                                                   Search: [          ]

Table: Inclusion Rate, Inclusion Proportion and Recruitment Rate by Center.

| Center | Included | Recruited | Total_Time | Inclusion Rate | Inclusion Proportion | Recruitment Rate |
|---|---|---|---|---|---|---|
| 02__crphf | 42 | 61 | 518 | 2.4324 | 0.6885 | 3.5328 |
| 03__ufms | 2 | 2 | 518 | 0.1158 | 1 | 0.1158 |
| 05__hmb | 41 | 65 | 518 | 2.3745 | 0.6308 | 3.7645 |
| 06__ufmg | 15 | 22 | 518 | 0.8687 | 0.6818 | 1.2741 |
| 07__hnb | 47 | 96 | 518 | 2.722 | 0.4896 | 5.5598 |
| 10__cepem | 23 | 31 | 518 | 1.332 | 0.7419 | 1.7954 |

Showing 1 to 6 of 6 entries                                    Previous   1   Next

Figure: Inclusion Proportion, Inclusion Rate and Recruitment Rate by Center.

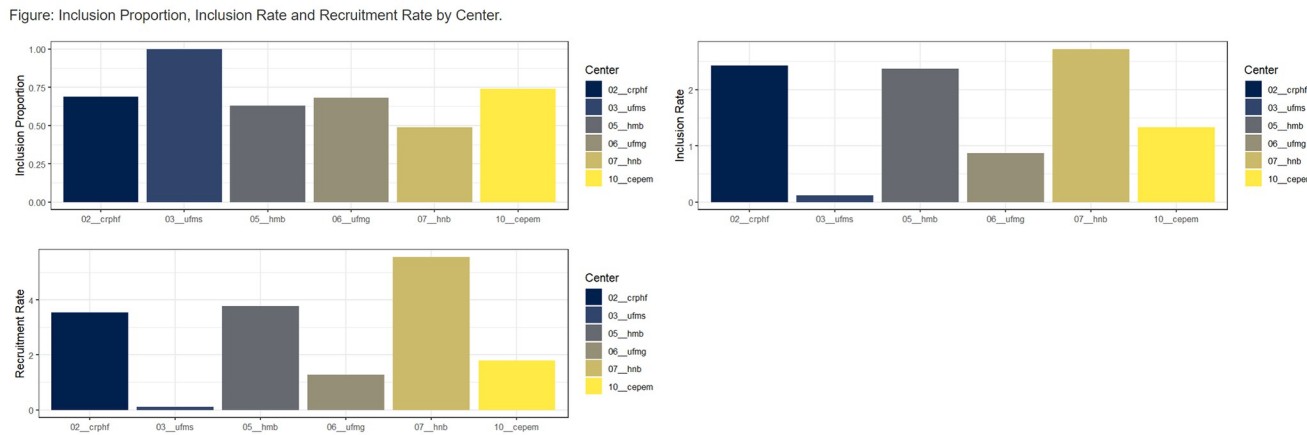

**Fig 2. Subjects' recruitment vs. inclusion enrolment rates per trial site.**

### Subjects' inclusion rates

Subjects' inclusion rates are the main indicator of trial performance. They allow for to optimisation of resource allocation and implement recruitment strategies if the rates are lower than expected. The patients' inclusion section (Figs 1 and 2) inform the absolute numbers of subjects recruited and enrolled, the study duration (days), and a set of four performance indicators of the overall trial and trial sites: recruitment and inclusion enrolment rates (number of subjects recruited or included per month), the proportion of enrolled subjects (number of enrolled subjects vs. recruited subjects) and current inclusion enrolment rate (number of an actual subjects enrolled vs. the expected ones). Additionally, the overall current inclusion enrolment rate and the trial performance indicators per trial site are presented in graphs.

### Screening failure and queries evaluations

The screening failure is a critical trial performance issue that might imply financial losses, as the trial sites spend staff work hours and trial supplies running this activity. Moreover, recording the reasons for screening failure allows reporting of the study population according to the

CONSORT diagram. It depicts the passage of subjects through the trial avoiding selection bias underreporting. The comparison of screening failure reasons between trial sites offers an opportunity to identify the need for additional training as these frequencies should be equally distributed among sites in a randomized clinical trial. Fig 3 presents the absolute frequency of failures per trial site, and a pie chart displays the relative frequency. Similarly, on Fig 4, the reasons for screening failure are displayed.

The site engagement might be also appraised through queries evaluations including the trial sites' query response performance. A dashboard tab informs the mean time to respond or close a query as well as the frequency of queries per study site. Discrepancies in the number of queries between trial sites may indicate the need for training in protocol procedures. The

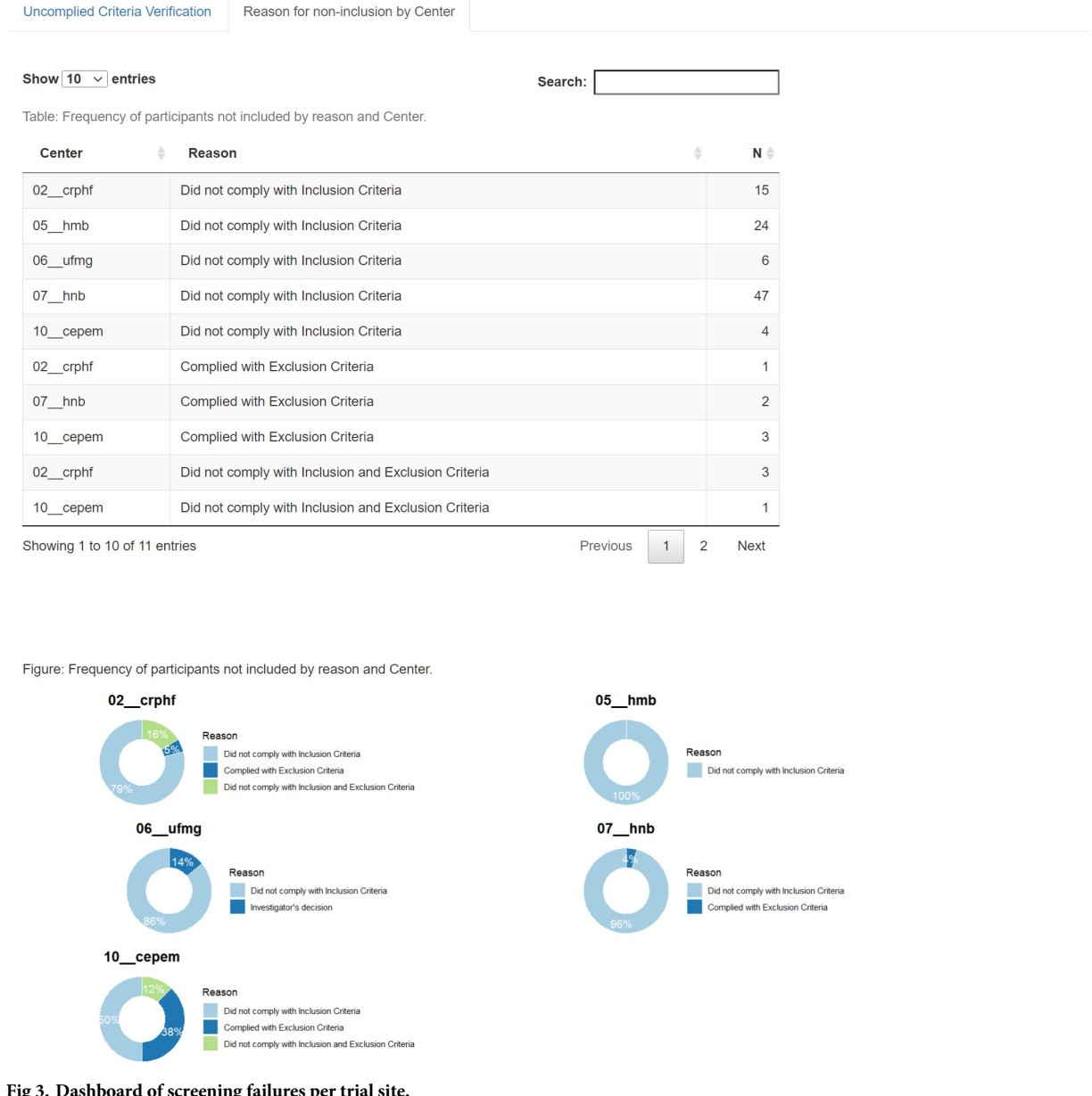

Fig 3. **Dashboard of screening failures per trial site.**

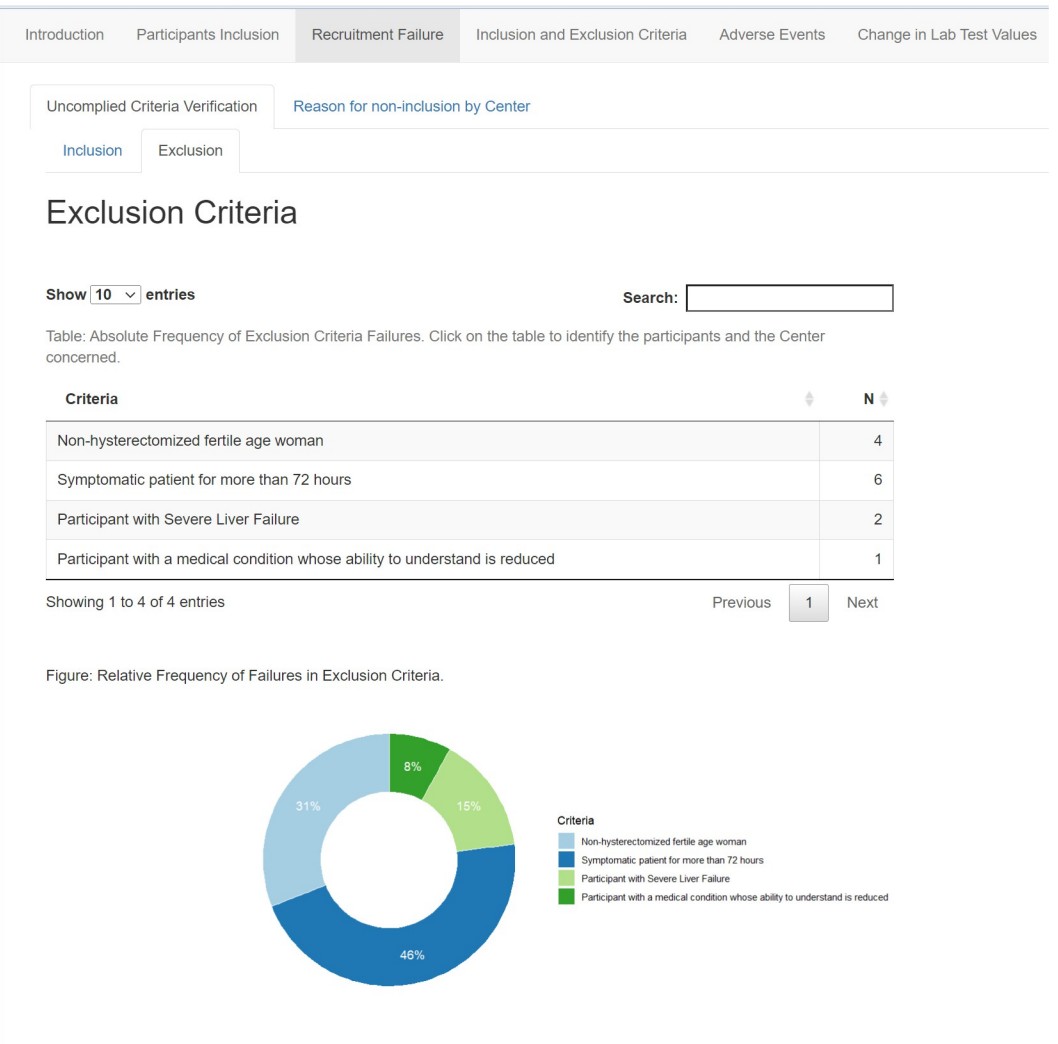

**Fig 4. Dashboard of reasons for screening failures.**

frequency of queries per variable is also important evidence that the trial site staff should be clarified on how to fill out the CRFs to avoid systematic errors.

### Follow-up visits according to the trial schedule

One of the most common protocol deviations is follow-up visits performed out of the time range defined in the protocols' study schedule. Adherence to the protocol can be measured by the percentage of follow-up visits around the due date range. The mean number of days outside the allowed time interval can also be described either per trial site or per subject's follow-up visit. As with every previous feature of this code, if a table line is selected, a new table is loaded underneath presenting the secondary ID of the subject (Fig 5). This allows the clinical research assistant (CRA) to open a query in the electronic data capture system promptly. Fig 5 shows the follow-up visit deviations per trial site, while Fig 6 shows the information per subject's visit.

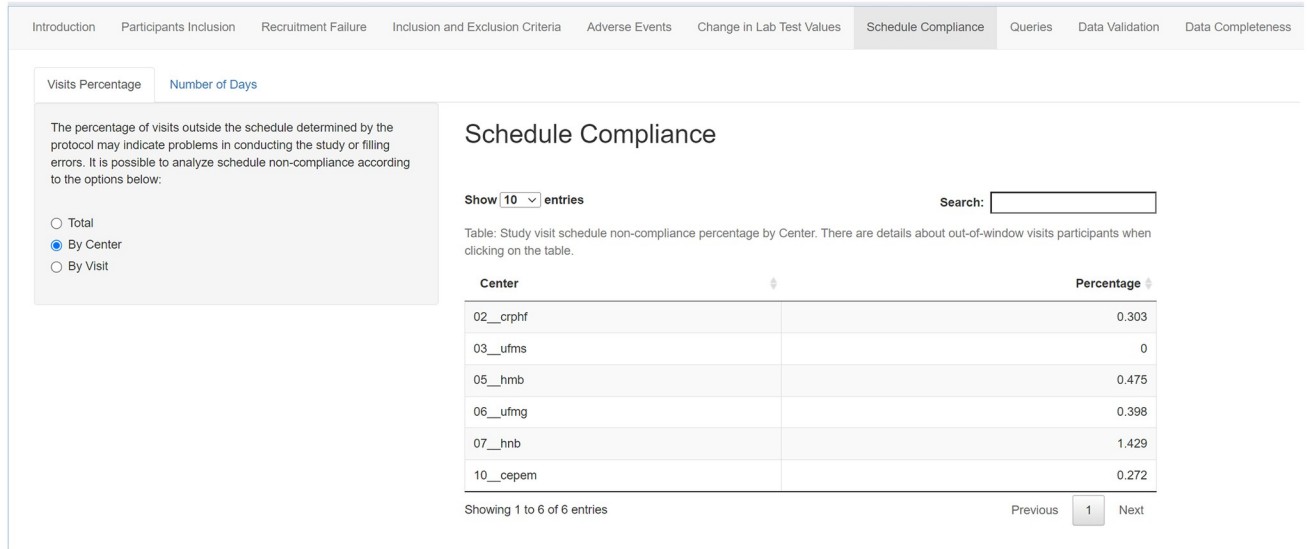

**Fig 5. Screenshot of follow-up visits evaluation per trial site.**

## Inclusion and exclusion criteria, adverse events and laboratory results

Including subjects outside the inclusion criteria age range is a protocol violation that must be avoided in order to ensure the safety of the trial subjects. Age and pregnancy are the sort of data that needs 100% SDV, however, this code adds an additional safety layer, allowing to

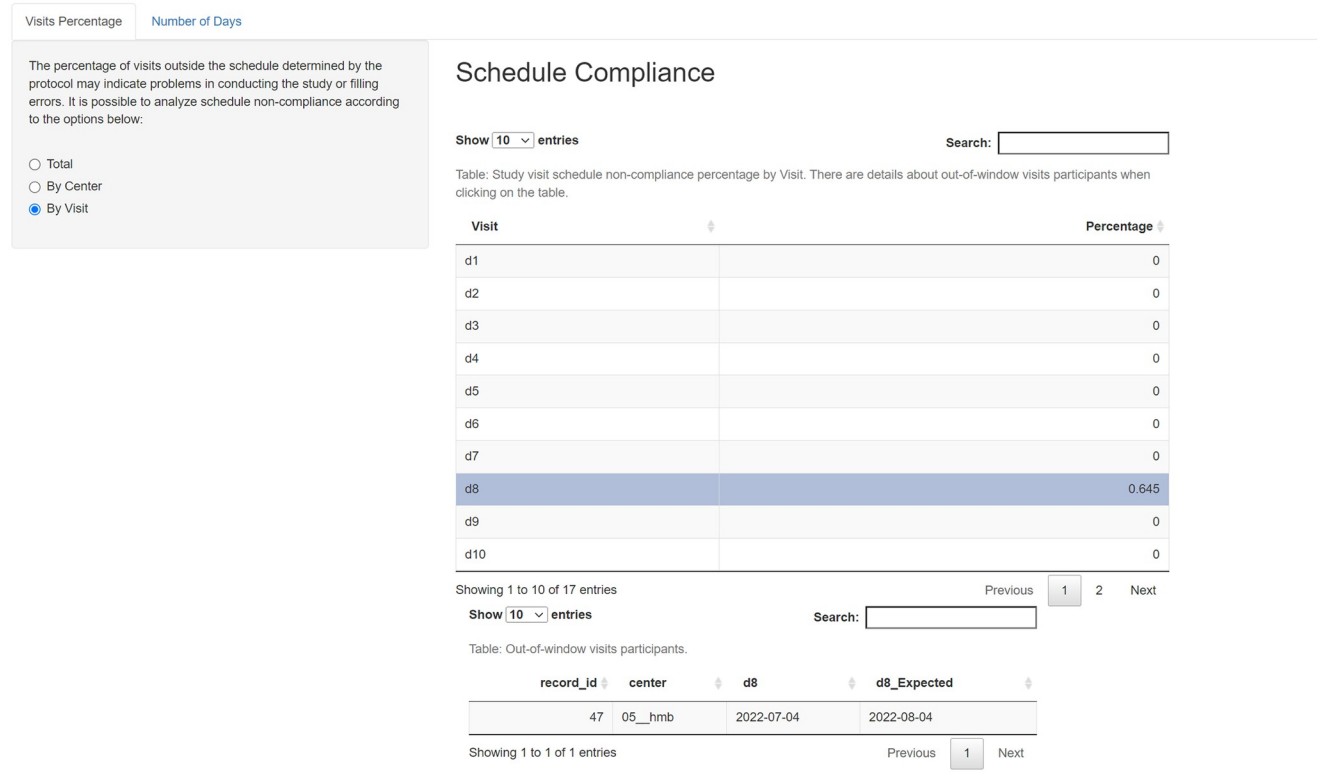

**Fig 6. Screenshot of follow-up visits evaluation per subject' visit.**

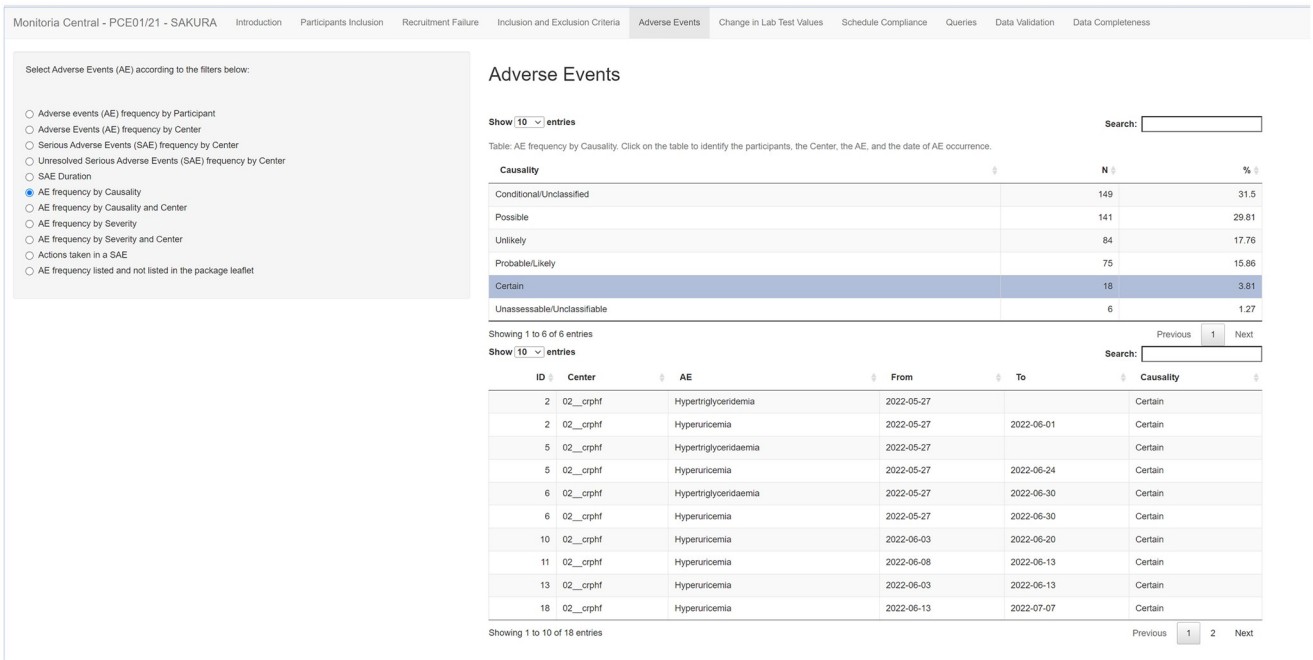

**Fig 7. AE dashboard: AE frequency and ID by causality.**

check remotely the difference between the date of birth and the date of inclusion enrolment, as well as pregnancy.

Similarly, pharmacovigilance safety flags cannot be missed. The dashboard allows monitoring all adverse events (AE) aggregated as follows: frequency of AE per subject and per study trial site; frequency of AE per causality in each trial site (Fig 7); frequency of AE per intensity in each trial site; frequency of unexpected or expected AE according to the drug factsheet; frequency of serious adverse event (SAE) per site; ongoing SAE per trial site and actions taken after the SAE. As per the previous code's features, a table of the secondary ID, trial site, and start and end dates are provided for each of these AE's evaluations. A search field allows further refinement of the AEs investigations. These are major parameters for the pharmacovigilance staff and may inform a Data Safety Monitoring Board (DSMB) report.

Laboratory results out of the reference value interval are an additional evaluation of AE that might be underreported. The dashboard presents the absolute and relative frequency of abnormal laboratory results by trial site, exam and subject follow-up visit. If a result is selected, a table listing the subjects' secondary ID and individual results is shown.

## Data quality checks: Validation and completeness

The dates validation tabs cross-check the inform consent (IC) signing date and the AE and concomitant medicine report date, as these reports cannot be produced before the IC signing date. The REDCap CRF's fields to report laboratory and physical exam results were designed to show warnings when an unexpected result is inputted (internal check). However, data entry is not blocked as these may be the real results. Discrepant values must be double-checked, though. The data validation feature allows a quick check of unlikely or out-of-range values for laboratory and physical exam results, such as blood pressure and weight.

Unchecked outcomes missing values may undermine sample size calculation reducing the power of results and utterly damaging the entire clinical trial result. Both, overall and per trial

site, safety and efficacy outcomes completeness can be checked at a glance using the outcomes missing values tab. Alike, the absence of missing values in the subject´s inclusion and exclusion criteria fields might preserve the integrity of the trial's internal validity as it ensures that the study population is in accordance with the study protocol.

## Discussion

While the Good Clinical Practice (GCP) core principles remain the same since it was first issued in 1996, when clinical research was largely paper-based, the means to ensure that the safety and well-being of human subjects participating in clinical trials are protected and that the results of the clinical trials are credible have evolved.

To avoid interruptions of the most needed therapeutic trials for COVID-19 and assure the quality of data according to GCP and regulatory requirements, the industries, contract research organizations (CRO) and regulatory agencies have been encouraged to transition to risk-based monitoring (RBM). Although RBM is not a response designed to mitigate the impact of COVID-19 on clinical trials, its adoption, which has been recommended for almost 10 years, was strongly needed in the context of the pandemic.

In June 2023, the U.S. Food and Drug Administration announced the availability of a draft guidance with updated recommendations for GCP aimed at modernizing the design and conduct of clinical trials, making them more agile without compromising data integrity or participant protections. The draft guidance is adopted from the International Council for Harmonisation's (ICH) recently updated E6(R3) draft guideline that was developed to enable the incorporation of rapidly developing technological and methodological innovations into the clinical trial enterprise [17], currently under public consultation.

The centralized monitoring tool for the remote appraisal of electronic data was designed to address the most common protocol deviations, pharmacovigilance alert signs, study performance and data completeness through automated evaluations. A script (R software) accesses the clinical trial dataset to perform these evaluations. None of these breaks the blind or information about the outcomes. The data are presented in dashboards, but, they cannot be modified using this tool. To preserve the trial dataset integrity and audit logs for tracking data handling, all discrepancies identified by the script must be manually corrected through the electronic data capture system (REDCap) query module.

There are many approaches to evaluating the quality of a clinical trial dataset, and although they are complementary, using R script-based central monitoring has advantages. As the R script is coded as a structured language, it assures the reproducibility of these assessments across actors and trials [18]. For instance, REDCap reports are more prone to errors as unreliable variables selection.

Cleaning the data before freezing the dataset for statistical analysis is a hard and time-consuming process [19]. Central monitoring does not replace this analysis step, but, in our experience, it accelerates it, as the data quality evaluation is performed throughout the trial. Similarly, the number of monitoring on-site visits could also be decreased, which has a positive impact on trial costs.

Recently, the ICH Harmonised Guideline Good Clinical Practice (GCP) E6(R3) ICH Consensus Guideline recognized that centralised monitoring processes provide additional monitoring capabilities that can complement and reduce the extent and/or frequency of site monitoring or be used on its own. Use of centralised data analytics can help identify systemic or site-specific issues, including protocol non-compliance and potentially unreliable data [17]. The main strength of the manuscript is that it presents an open-science solution to push

further in the centralized monitoring adoption direction, as these tools' implementation will be reinforced by this regulatory guidance soon.

These real-time data evaluations also enhance the subject's safety, as the AEs control is improved [20]. The tool could also facilitate the DSMB procedures, meetings presentations and reports. Additionally, online access can be granted to DSMB members allowing them to perform safety overviews anytime. The real-time trial site performance metrics also helped driving study coordination informed decisions, such as the need for new recruitment strategies, pieces of training or even interruption of trial sites.

This version of the code has limitations. The results are descriptive and, in large trials, further statistical tests may be needed [21]. There is no restricted access per role within the trial, as for most clinical data software. The query assessments use data from a REDCap audit log, meaning that a routine to download this dataset is needed to update the site performance metrics dashboard. Still, the inclusion of trial sites alongside the trial requires adjustments to the code to reproduce the actual expected recruitment rate. Finally, as the tool was developed using the REDCap data capture system and long-format datasets, additional work is needed to run the script in other formats or systems. However, as it is an open source software, future improvements of this open-science solution through collaborations and further data sharing are expected [22]. A future proposal for open-science solutions includes sharing CRFs developed using CDISC for neglected tropical diseases clinical networks, allowing the use of the same centralized monitoring scripts in distinct trials. Harmonizing the data quality and datasets might promote individual data pooled metanalyses speeding up the synthesis of evidence where scarce data are generated.

## Conclusion

Sharing data between researchers worldwide is an increasingly important aspect of addressing diseases and developing new therapeutic, diagnostic, and epidemiological approaches. Faced with the perspective of ensuring sustainability in Public Institutions, it is important to consider free methodologies and tools that support greater access and execution of quality research. These open science solutions are a complementary tool to clinical trials' data management and monitoring teams rather than an on-site monitoring substitute. Together with other open science initiatives, these powerful tools may help push towards the adoption of centralized monitoring and RBM. Centralized monitoring is an additional data quality assurance layer that might be regarded as a mandatory quality standard in clinical trials in a near future.

## Author Contributions

**Conceptualization:** André Daher, Júlio Castro-Alves.

**Data curation:** Júlio Castro-Alves, Karla Regina Gram dos Santos, Cristiane Siqueira do Valle, Margareth Catoia Varela.

**Funding acquisition:** André Daher, Rodrigo Correa Oliveira.

**Methodology:** Júlio Castro-Alves.

**Project administration:** André Daher.

**Software:** Júlio Castro-Alves, Leandro Amparo, Natalia Pacheco de Moraes, Thaís Regina Araújo dos Santos, Margareth Catoia Varela.

**Supervision:** André Daher.

**Validation:** Rodrigo Correa Oliveira.

**Writing – original draft:** André Daher.

**Writing – review & editing:** Júlio Castro-Alves, Maria Hermoso, Margareth Catoia Varela, Rodrigo Correa Oliveira.

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
