## [Decision Letter · Decision Letter 0]

14 Aug 2023

PONE-D-23-18254A code for clinical trials centralized monitoring, sharing open-science solutions to high-quality dataPLOS ONE

Dear Dr. Hermoso,

Thank you for submitting your manuscript to PLOS ONE. After careful consideration, we feel that it has merit but does not fully meet PLOS ONE’s publication criteria as it currently stands. Therefore, we invite you to submit a revised version of the manuscript that addresses the points raised during the review process.

We look forward to receiving your revised manuscript.

Kind regards,

Claudia Garcia Serpa Osorio-de-Castro, Ph.D

Academic Editor

PLOS ONE

“The Vice-presidency of Research and Biological Collections, Oswaldo Cruz Foundation

(Fiocruz), Ministry of Health of Brazil provided the funding.”

6. Please upload a new copy of Figures 1-4 as the detail is not clear. Please follow the link for more information: https://blogs.plos.org/plos/2019/06/looking-good-tips-for-creating-your-plos-figures-graphics/" https://blogs.plos.org/plos/2019/06/looking-good-tips-for-creating-your-plos-figures-graphics/

Reviewers' comments:

Reviewer's Responses to Questions

**Comments to the Author**

1. Is the manuscript technically sound, and do the data support the conclusions?

Reviewer #1: Yes

Reviewer #2: Yes

2. Has the statistical analysis been performed appropriately and rigorously? 

Reviewer #1: Yes

Reviewer #2: N/A

3. Have the authors made all data underlying the findings in their manuscript fully available?

Reviewer #1: Yes

Reviewer #2: Yes

4. Is the manuscript presented in an intelligible fashion and written in standard English?

Reviewer #1: Yes

Reviewer #2: Yes

5. Review Comments to the Author

Reviewer #1: The article present an enthusiastic opportunity for optimizing clinical trials timelines.

Please, consider some revisions and corrections below:

Line 122: Include details about the location of the study. Was it only Brazilian patients?

Line 191: The sentence is indicating that all the trial CRF are available upon request. Please indicate if there are restrictions or protections on the information provideded.

Line 196: Make clear if the available information will be presented only in Portuguese.

Line 303: Include a rational to explain why the parameters that were chosen to be automated evaluated are representing the most critical aspects of monitoring. Are those representative to the critical aspects to be monitored?

Line 313: Provide an improved explanation for this statement.

Line 321: Include a discussion on how the presentation of the tool only in Portuguese represents a limitation on access?

Line 326: Why and which "further statistical tests" will be needed?

Reviewer #2: This is a manuscript on a relevant topic, the monitoring of clinical trials that is critical both for the protection of human subjects and for the conduct of high-quality research. It is written in a scientifically clear and adequate manner, but some minor adjustments are needed such as definitions of acronyms and some details in the discussion section.

6. PLOS authors have the option to publish the peer review history of their article (what does this mean?). If published, this will include your full peer review and any attached files.

Reviewer #1: **Yes: **Gustavo Mendes Lima Santos

Reviewer #2: **Yes: **CARLOS AUGUSTO FERREIRA DE ANDRADE

---

## [Author Response · Author response to Decision Letter 0]

29 Sep 2023

Please see also rebuttal letter. 

We are grateful for the revision of publication criteria and for the reviewer’s comments and suggestions. Please find below our responses to the reviewers. 

Reviewer #1: The article present an enthusiastic opportunity for optimizing clinical trials timelines.

Please, consider some revisions and corrections below:

Line 122: Include details about the location of the study. Was it only Brazilian patients?

The following text was added (line 122): The trial was performed in six clinical trial sites in Brazil, located in 5 different states (Rio de Janeiro, São Paulo, Minas Gerais, Rondônia, and Mato Grosso do Sul).

Line 191: The sentence is indicating that all the trial CRF are available upon request. Please indicate if there are restrictions or protections on the information provided.

The code is licensed under the Attribution-NonCommercial 4.0 International (CC BY-NC 4.0). The follow statement was inserted (line 187): “These files are licensed under a Creative commons Attribution-Non-commercial 4.0 International (CC BY-NC 4.0)”

Line 196: Make clear if the available information will be presented only in Portuguese.

In fact, currently the dashboard shows the information only in Portuguese. This information has been added to the text (line 205)

Line 303: Include a rational to explain why the parameters that were chosen to be automated evaluated are representing the most critical aspects of monitoring. Are those representative to the critical aspects to be monitored?

You are correct, the chosen parameters are critical monitoring aspects. For the sake of clarification, the following statement was inserted in line 198: 

“Most of the critical central monitoring aspects are evaluated. The rational of parameters election were based on the most frequent monitoring findings, i.e protocol deviations; data robustness and integrity (completeness of outcomes); and high risk aspects, as pharmacovigilance. Additionally, study sites performance metrics were included as alert signs.” 

Line 313: Provide an improved explanation for this statement.

The statements was rephrased to: As the R script is coded as a structured language, it assures the reproducibility of these assessments across actors and trials [17]. For instance, REDCap reports are more prone to errors during the variables selection (line 324).

Line 321: Include a discussion on how the presentation of the tool only in Portuguese represents a limitation on access?

The following sentence was inserted at line 346: “Currently, the dashboards are presented in Portuguese, although this language barrier may be overcome by collaborative science.”

Line 326: Why and which "further statistical tests" will be needed?

The following reference was included (line 348): A statistical approach to central monitoring of data quality in clinical trials D Venet, E Doffagne, T Burzykowski, F Beckers, Y Tellier, E Genevois-Marlin, U Becker…Clinical Trials, 2012.

Reviewer #2: This is a manuscript on a relevant topic, the monitoring of clinical trials that is critical both for the protection of human subjects and for the conduct of high-quality research. It is written in a scientifically clear and adequate manner, but some minor adjustments are needed such as definitions of acronyms and some details in the discussion section.

This reviewer’s comments were included in the file named “CARLOS-SUG-PONE-D-23-18254”

• Definition of acronyms have been included in the revised manuscript:

o Line 27: Case Report Forms (CRFs)

o Line 181: Structured Query Language (SQL)

o Line 296: Good Clinical Practice (GCP)

• Link to the dashboard has been checked for proper functioning (line 198): https://shiny.fiocruz.br/teste/pce0121/

• Regarding the text of figures, which the reviewer suggests to translate into English. Unfortunately translating the figures into English are not possible. The figures were extracted from the site, in order to translate the figures, we must translate the code. This is a limitation that has been highlighted in the manuscript.

• In the discussion section, an explicit presentation of the strengths of the manuscript, as well as future proposals for open-science solutions was added as suggested by the reviewer: line 337.

• We have included an important update on the clinical trials regulatory landscape: the new version of the Good Clinical Practice (ICH E6 R3), the first review in the last 20 years. This paper presents an open-science solution to the very edge of this new landscape of regulatory requirements. The following paragraph was inserted (line 332):

“Recently, the ICH Harmonised Guideline Good Clinical Practice (GCP) E6(R3) ICH Consensus Guideline recognized that centralised monitoring processes provide additional monitoring capabilities that can complement and reduce the extent and/or frequency of site monitoring or be used on its own. Use of centralised data analytics can help identify systemic or site-specific issues, including protocol non-compliance and potentially unreliable data (ref). The main strength of the manuscript is that it presents an open-science solution to push further in the centralized monitoring adoption direction, as these tools' implementation will be reinforced by this regulatory guidance soon.”

Finally in order to inform future proposals for open-science solutions we added a last paragraph at the review (line 357): 

“A future proposal for open-science solutions includes sharing CRFs developed using CDISC for neglected tropical diseases clinical networks, allowing the use of the same centralized monitoring scripts in distinct trials. Harmonizing the data quality and datasets might promote individual data pooled metanalyses speeding up the synthesis of evidence where scarce data are generated.”

Still in the discussion section, the reviewer suggests including some references in two paragraphs (lines 315 to 335 in the original manuscript).

• We have included the reference: Risk-based centralized data monitoring of clinical trials at the time of COVID-19 pandemic Most Alina Afroz a, Grant Schwarber b, Mohammad Alfrad Nobel Bhuiyan b,. Barnes B, Stansbury N, Brown D, Garson L, Gerard G, Piccoli N, et al. Risk-based monitoring in clinical trials: past, present, and future. Therapeutic innovation & regulatory science. 2021;55: 899–906 Additionally, we have provided a relevant reference by the ICH Consensus Guideline (GCP E6(R3)), currently under public consultation. However, the statements are based on reported experience by using the central monitoring tool. We hope that further references will be available in the future reporting the use of similar tools. We have rephrased the two paragraphs to make this clear.

---

## [Editor Report · Decision Letter 1]

5 Oct 2023

PONE-D-23-18254R1A code for clinical trials centralized monitoring, sharing open-science solutions to high-quality dataPLOS ONE

Dear Dr. Hermoso Thank you for submitting your manuscript to PLOS ONE. After careful consideration, we feel that it has merit but does not fully meet PLOS ONE’s publication criteria as it currently stands. Therefore, we invite you to submit a revised version of the manuscript that addresses the points raised during the review process. Most of the Reviewers´suggestions and recommendations were met.

However, all figures and captions are in Portuguese and quality of these elements is still not adequate.

Portuguese-language in Figures was highlighted by a reviewer and the answer was: "Regarding the text of figures, which the reviewer suggests to translate into English. Unfortunately translating the figures into English are not possible. The figures were extracted from the site, in order to translate the figures, we must translate the code. This is a limitation that has been highlighted in the manuscript."

Publishing in English is a standard procedure and if figures are generated by code, this must be thought of in advance. It would be wise to translate the code and generate figures in English as per PlosOne policy (PlosOne Publication Criterium #5 states that "The article is presented in an intelligible fashion and is written in standard English". Although figures are intelligible, they are so only for Portuguese-Language speakers.

We look forward to receiving your revised manuscript.

Kind regards,

Claudia Garcia Serpa Osorio-de-Castro, Ph.D

Academic Editor

PLOS ONE
---

## [Author Response · Author response to Decision Letter 1]

27 Oct 2023

We are grateful for the revision of publication criteria and for the reviewer’s comments and suggestions. Please find below our responses to the reviewers. 

Reviewer #1: The article present an enthusiastic opportunity for optimizing clinical trials timelines.

Please, consider some revisions and corrections below:

Line 122: Include details about the location of the study. Was it only Brazilian patients?

The following text was added (line 122): The trial was performed in six clinical trial sites in Brazil, located in 5 different states (Rio de Janeiro, São Paulo, Minas Gerais, Rondônia, and Mato Grosso do Sul).

Line 191: The sentence is indicating that all the trial CRF are available upon request. Please indicate if there are restrictions or protections on the information provided.

The code is licensed under the Attribution-NonCommercial 4.0 International (CC BY-NC 4.0). The follow statement was inserted (line 187): “These files are licensed under a Creative commons Attribution-Non-commercial 4.0 International (CC BY-NC 4.0)”

Line 196: Make clear if the available information will be presented only in Portuguese.

In fact, currently the dashboard shows the information only in Portuguese. This information has been added to the text (line 205)

Line 303: Include a rational to explain why the parameters that were chosen to be automated evaluated are representing the most critical aspects of monitoring. Are those representative to the critical aspects to be monitored?

You are correct, the chosen parameters are critical monitoring aspects. For the sake of clarification, the following statement was inserted in line 198: 

“Most of the critical central monitoring aspects are evaluated. The rational of parameters election were based on the most frequent monitoring findings, i.e protocol deviations; data robustness and integrity (completeness of outcomes); and high risk aspects, as pharmacovigilance. Additionally, study sites performance metrics were included as alert signs.” 

Line 313: Provide an improved explanation for this statement.

The statements was rephrased to: As the R script is coded as a structured language, it assures the reproducibility of these assessments across actors and trials [17]. For instance, REDCap reports are more prone to errors during the variables selection (line 324).

Line 321: Include a discussion on how the presentation of the tool only in Portuguese represents a limitation on access?

The following sentence was inserted at line 346: “Currently, the dashboards are presented in Portuguese, although this language barrier may be overcome by collaborative science.”

Line 326: Why and which "further statistical tests" will be needed?

The following reference was included (line 348): A statistical approach to central monitoring of data quality in clinical trials D Venet, E Doffagne, T Burzykowski, F Beckers, Y Tellier, E Genevois-Marlin, U Becker…Clinical Trials, 2012.

Reviewer #2: This is a manuscript on a relevant topic, the monitoring of clinical trials that is critical both for the protection of human subjects and for the conduct of high-quality research. It is written in a scientifically clear and adequate manner, but some minor adjustments are needed such as definitions of acronyms and some details in the discussion section.

This reviewer’s comments were included in the file named “CARLOS-SUG-PONE-D-23-18254”

• Definition of acronyms have been included in the revised manuscript:

o Line 27: Case Report Forms (CRFs)

o Line 181: Structured Query Language (SQL)

o Line 296: Good Clinical Practice (GCP)

• Link to the dashboard has been checked for proper functioning (line 198): https://shiny.fiocruz.br/teste/pce0121/

• All figures have been translated into English (figures 1 to 7).

• In the discussion section, an explicit presentation of the strengths of the manuscript, as well as future proposals for open-science solutions was added as suggested by the reviewer: line 337.

• We have included an important update on the clinical trials regulatory landscape: the new version of the Good Clinical Practice (ICH E6 R3), the first review in the last 20 years. This paper presents an open-science solution to the very edge of this new landscape of regulatory requirements. The following paragraph was inserted (line 332):

“Recently, the ICH Harmonised Guideline Good Clinical Practice (GCP) E6(R3) ICH Consensus Guideline recognized that centralised monitoring processes provide additional monitoring capabilities that can complement and reduce the extent and/or frequency of site monitoring or be used on its own. Use of centralised data analytics can help identify systemic or site-specific issues, including protocol non-compliance and potentially unreliable data (ref). The main strength of the manuscript is that it presents an open-science solution to push further in the centralized monitoring adoption direction, as these tools' implementation will be reinforced by this regulatory guidance soon.”

Finally in order to inform future proposals for open-science solutions we added a last paragraph at the review (line 357): 

“A future proposal for open-science solutions includes sharing CRFs developed using CDISC for neglected tropical diseases clinical networks, allowing the use of the same centralized monitoring scripts in distinct trials. Harmonizing the data quality and datasets might promote individual data pooled metanalyses speeding up the synthesis of evidence where scarce data are generated.”

Still in the discussion section, the reviewer suggests including some references in two paragraphs (lines 315 to 335 in the original manuscript).

• We have included the reference: Risk-based centralized data monitoring of clinical trials at the time of COVID-19 pandemic Most Alina Afroz a, Grant Schwarber b, Mohammad Alfrad Nobel Bhuiyan b,. Barnes B, Stansbury N, Brown D, Garson L, Gerard G, Piccoli N, et al. Risk-based monitoring in clinical trials: past, present, and future. Therapeutic innovation & regulatory science. 2021;55: 899–906 Additionally, we have provided a relevant reference by the ICH Consensus Guideline (GCP E6(R3)), currently under public consultation. However, the statements are based on reported experience by using the central monitoring tool. We hope that further references will be available in the future reporting the use of similar tools. We have rephrased the two paragraphs to make this clear.

---

## [Editor Report · Decision Letter 2]

31 Oct 2023

A code for clinical trials centralized monitoring, sharing open-science solutions to high-quality data

PONE-D-23-18254R2

Dear Dr. Hermoso,

We’re pleased to inform you that your manuscript has been judged scientifically suitable for publication and will be formally accepted for publication once it meets all outstanding technical requirements.

Kind regards,

Claudia Garcia Serpa Osorio-de-Castro, Ph.D

Academic Editor

PLOS ONE

Additional Editor Comments (optional):

The graphic elements are now all in English, meeting PlosOne requirements.

---

## [Editor Report · Acceptance letter]

7 Nov 2023

PONE-D-23-18254R2 

A code for clinical trials centralized monitoring, *sharing open-science solutions to high-quality data*

Dear Dr. Hermoso:

I'm pleased to inform you that your manuscript has been deemed suitable for publication in PLOS ONE. Congratulations! Your manuscript is now with our production department. 

Kind regards, 

on behalf of

Dr. Claudia Garcia Serpa Osorio-de-Castro 

Academic Editor

PLOS ONE